# Direct RNA Nanopore Sequencing of SARS-CoV-2 Extracted from Critical Material from Swabs

**DOI:** 10.3390/life12010069

**Published:** 2022-01-04

**Authors:** Davide Vacca, Antonino Fiannaca, Fabio Tramuto, Valeria Cancila, Laura La Paglia, Walter Mazzucco, Alessandro Gulino, Massimo La Rosa, Carmelo Massimo Maida, Gaia Morello, Beatrice Belmonte, Alessandra Casuccio, Rosario Maugeri, Gerardo Iacopino, Carmela Rita Balistreri, Francesco Vitale, Claudio Tripodo, Alfonso Urso

**Affiliations:** 1Department of Surgical, Oncological and Oral Sciences, University of Palermo, 90127 Palermo, Italy; 2CNR-ICAR, National Research Council of Italy, Via Ugo La Malfa, a5c, 90146 Palermo, Italy; antonino.fiannaca@icar.cnr.it (A.F.); laura.lapaglia@icar.cnr.it (L.L.P.); massimo.larosa@icar.cnr.it (M.L.R.); alfonso.urso@icar.cnr.it (A.U.); 3Department of Health Promotion, Mother and Child Care, Internal Medicine and Medical Specialties “G. D’Alessandro”, Hygiene Section, University of Palermo, 90127 Palermo, Italy; fabio.tramuto@unipa.it (F.T.); walter.mazzucco@unipa.it (W.M.); carmelo.maida@unipa.it (C.M.M.); alessandra.casuccio@unipa.it (A.C.); francesco.vitale@unipa.it (F.V.); 4Tumor Immunology Unit, Department of Health Promotion, Mother and Child Care, Internal Medicine and Medical Specialties “G. D’Alessandro”, University of Palermo, 90127 Palermo, Italy; valeria.cancila@unipa.it (V.C.); gaia.morello@unipa.it (G.M.); beatrice.belmonte@unipa.it (B.B.); claudio.tripodo@unipa.it (C.T.); 5Cogentech srl Società Benefit, FIRC Institute of Molecular Oncology (IFOM), Via Adamello 16, 20139 Milan, Italy; alessandro.gulino@cogentech.it; 6Department of Experimental Biomedicine and Clinical Neurosciences, School of Medicine, Neurosurgical Clinic, University of Palermo, 90127 Palermo, Italy; rosario.maugeri1977@gmail.com (R.M.); gerardo.iacopino@gmail.com (G.I.); 7Department of Biomedicine, Neuroscience and Advanced Diagnostics (Bi.N.D.), University of Palermo, 90134 Palermo, Italy; carmelarita.balistreri@unipa.it

**Keywords:** MinION, direct RNA nanopore sequencing, SARS-CoV-2, COVID-19, swab

## Abstract

In consideration of the increasing prevalence of COVID-19 cases in several countries and the resulting demand for unbiased sequencing approaches, we performed a direct RNA sequencing (direct RNA seq.) experiment using critical oropharyngeal swab samples collected from Italian patients infected with SARS-CoV-2 from the Palermo region in Sicily. Here, we identified the sequences SARS-CoV-2 directly in RNA extracted from critical samples using the Oxford Nanopore MinION technology without prior cDNA retrotranscription. Using an appropriate bioinformatics pipeline, we could identify mutations in the nucleocapsid (N) gene, which have been reported previously in studies conducted in other countries. In conclusion, to the best of our knowledge, the technique used in this study has not been used for SARS-CoV-2 detection previously owing to the difficulties in the extraction of RNA of sufficient quantity and quality from routine oropharyngeal swabs. Despite these limitations, this approach provides the advantages of true native RNA sequencing and does not include amplification steps that could introduce systematic errors. This study can provide novel information relevant to the current strategies adopted in SARS-CoV-2 next-generation sequencing.

## 1. Introduction

Currently, the characterization of the SARS-CoV-2 genome has become a priority for global healthcare in order to identify more suitable vaccines and therapeutic drugs [1]. The use of third-generation sequencing technologies has significantly increased in recent years, because these methods yield reliably long reads even from biological samples with noise [2]. In addition, the use of swab samples for RNA extraction shows critical issues, regarding both the fragmentation level and concentration. Such limitations affect the read coverage of sequencing, which is decreased in a considerable manner. Protocols used for RNA library construction often require at least 100 ng total RNA. However, under certain conditions, it is not possible to obtain that much RNA to satisfy this requirement. In these cases, in order to increase RNA amount, several protocols suggest RNA enrichment by reverse transcription and amplification of the cDNA, prior to library preparation [3].

Even though cDNA sequencing is considered the gold standard for the analysis of critical materials [3], DNA polymerase may introduce certain read errors during the DNA synthesis step, which can considerably affect the analysis [4,5].

For directly sequencing the coronavirus genome from clinical samples, such as nasopharyngeal or oropharyngeal swabs, the suggested protocol by Oxford Nanopore is the PCR tailing of SARS-CoV-2 [6]. It has been developed by the Artic Network for the sequencing of Ebola, Zika, and Chikungunya genomes [7,8]. Consequently, it was promptly adjusted for rapid sequence determination of SARS-CoV-2 RNA, in January 2020 [9]. Additional studies have led to improvements of this protocol, such as the simplification of the sequencing library preparation and increased sample multiplexing (up to 96), which has brought down the costs of the sequencing for a single sample to about GBP 10 [9]. In this way, this approach has become affordable for epidemiologic surveillance of the pandemic [10].

The PCR tailing of the SARS-CoV-2 protocol is based on a multiplex primers scheme that allows coverage of the whole genome. However, sequencing protocols taking advantage of target enrichment through the primers set, such as the PCR tailing of SARS-CoV-2, might be difficult in recovering individual mutations in underrepresented amplicons. This aspect may be considered relevant for the exceptional genetic plasticity of the genomes of RNA viruses. Indeed, the high mutation and recombination rates during genome replication by viral RNA-dependent RNA polymerases lead to populations of closely related viruses, so-called “quasispecies.” [11].

Conversely, the direct RNA sequencing (direct RNA seq.) of SARS-CoV-2 has the advantage of displaying its native sequence, Figure 1. In fact, the direct RNA seq. (SQK-RNA002) protocol enables exploring attributes of native RNA without contamination by artifacts originating from the in vitro cDNA amplification step, which is often necessary for target enrichment [4,5].

In this study, we investigated the suitability of Nanopore Oxford MinION Mk1B [12], a third-generation nanopore-based platform, for the identification of a critical target such as SARS-CoV-2 native RNA from a swab sample and, at the same time, tested its sensitivity in the characterization of the viral mutational landscape. Finally, we analyzed RNA samples extracted from oropharyngeal swabs of ten patients with COVID-19 and sequenced them in two pools following the direct RNA seq. protocol SQK-RNA002, Nanopore Technologies [13]. Although the concentration of the library loaded in the sequencing flow cell was approximately 200 times lower than that required for the protocol, we clearly identified two mutations in the nucleocapsid phosphoprotein (N) gene with respect to the sequence of the strain isolated in Wuhan, which has been described in the literature [14].

## 2. Materials and Methods

After the collection of ten oropharyngeal swab samples from Italian patients with COVID-19, we assayed the concentration and fragmentation level of the extracted RNA to arrange it for sequencing, as recommended by Oxford Nanopore in the Input DNA/RNA quality control guidelines [15]. Next, we prepared two sequencing libraries from two distinct pools, as described below, and launched a computational pipeline for aligning the obtained reads. Lastly, we confirmed the mutation frequency in our sample either by Sanger sequencing or by real-time PCR (qPCR) assays.

### 2.1. Input RNA Collection and Quality Control Steps

Samples. For this study, we collected RNA samples from ten Sicilian patients between 19 and 23 March, 2020. The patients tested positive in the 2019-Novel Coronavirus (2019-nCoV) Real-Time rRT-PCR Panel (Centers for Disease Control and Prevention (CDC) Atlanta, GA 30333, USA). Furthermore, to compare the nanopore sequencing results, we included two samples from uninfected individuals as negative controls. In order to assay the suitability of our custom primers and to avoid possible PCR artifacts derived from high fragmented and very low concentrated RNA, such as that extracted from the swab, we used negative controls of high-quality RNA extracted from brain biopsies of two patients who underwent surgery before the beginning of the epidemic in Italy. The biopsies were made available from the Department of Experimental Biomedicine and Clinical Neurosciences, School of Medicine, Neurosurgical Clinic, University of Palermo.

RNA extraction. The swab buffer kit was assayed using the QIAamp Viral RNA Mini Kit in Automated purification of RNA on QIAcube Instruments (Qiagen, Hilden, Germany; Cat. No. 9001292) according to the manufacturer’s instructions.

Assessment of concentration and fragmentation level. The quality of the fragmented sample was assessed with an Agilent Bioanalyzer 2100 (Agilent, GA, USA; Cat. No. G2939BA) using the Agilent RNA 6000 Pico assay (Agilent, GA, USA; Cat. No. 5067-1513). We assayed 1 µL of each pure sample. The RNA extracted from the samples showed high fragmentation and the lowest concentration levels with an RIN index between 2.6 and 2.1 and concentration between 19 and 829 pg/µL.

### 2.2. Preparation of the Libraries and the Computational Pipeline

The samples were organized in two pools (A and B), each with a final volume of 10 µL. Pool A included samples from three patients: two with a PCR cycle threshold (Ct) value of 18 and one with a Ct value of 20. To elaborate, we pooled 8 µL of samples from the first two patients (4 µL from each) and 2 µL from the last patient. Pool B included RNA samples collected from ten patients (1 µL from each patient): three samples were common in Pool A, while the other seven had decreasing Ct values and were collected from two patients with a Ct value of 21, two with 22, two with 23, and one with 24.

Library preparation, priming, and commencement of a sequencing run. Both pools were sequenced using a MinION Mk1B sequencer with the SQK-RNA002 protocol. After preparing the sequencing libraries A and B, as previously described, to ligate retrotranscription adapters (RTA) to the 3′ end of the RNA molecules, we combined each library with 6 µL of a mix comprising 3 µL of NebNext Quick Ligation Reaction Buffer, 1.5 µL of T4 DNA Ligase, 1 µL of RTA, and 0.5 µL of 110 nM RNA CS (RCS). Each reaction was incubated for 10 min at 22 °C. Next, the reactants were mixed with 9 µL of nuclease-free water, 8 µL of 5× first-strand buffer, 4 µL of 0.1 DTT, and 2 µL of 10 mM dNTPs. For cDNA synthesis, we added 2 µL of SuperScript III, following which both reactions were incubated under the following conditions: 50 °C for 50 min, 70 °C for 10 min, and 4 °C in Mastercycler Pro S (Eppendorf, Hamburg, Germany, Order No. 6325EJ921257) before proceeding to the next step.

The subsequent wash step was performed according to the procedure for the specified protocol.

Next, we measured the concentration of 1 µL of each library using a Qubit 3.0 Fluorimeter and the dsDNA HS assay kit (Life Technologies, Cat. No. Q32851). The concentrations were 1.2 ng/µL and 0.6 ng/µL for Libraries A and B, respectively. Next, we performed the attachment of the 1D sequencing adapter step according to the manufacturer’s instructions and recorded final concentrations of 0.9 ng/µL and 0.3 ng/µL for A and B, respectively. Lastly, 20 µL of sample from each library was mixed with 17.5 µL of nuclease-free water and 37.5 µL of RNA running buffer (RRB) in a final volume of 75 µL. The assessment of the R9.4 Flow Cell, which was performed prior to the loading of the libraries, revealed that 1575 active pores were available for sequencing. We performed a new MinKNOW [16] (v19.12.5) experiment using only the A loading mix in the flow cell.

In MinKNOW we selected the SQK-RNA002 Kit and continued the sequencing process until approximately 80% of pores were available (i.e., after 4 h). At this point, we paused the process and loaded the B loading mix in the flow cell. Next, the run was resumed and continued till the number of reads generated was unvaried at 397 k (i.e., after approximately 27 h). At the end of the sequencing process, we obtained the “Fast5” raw data files.

Computational pipeline**.** We developed a computational pipeline to analyze the output from the Oxford Nanopore MinION device. The first step in the process involved the conversion of “Fast5” in “Fastq” format. For this, we used the GPU version of the Nanopore Guppy base caller (v3.4.4) tool [17] with the following parameters: “flow cell = FLO-MIN106” and “kit = SQK-RNA002”.

As a second step, we performed read quality control using the PycoQC (v2.5.0.21) software [18] with standard parameters. This tool computes metrics and generates multiple quality plots for nanopore technologies that allow the initial evaluation of the sequenced reads.

PycoQC can subsequently provide an overview of the overall quality of the reads. To clean the input data, we filtered the quality and read length using the NanoFilt (v2.7.0) tool [19]. This tool can also analyze the sequencing summary file generated by guppy_basecaller to refine the filtering process. For these reasons, we filtered the reads using the sequencing summary file under the following parameters: minimum read length ≥500 nt and read quality ≥8.

Thereafter, only the filtered reads were considered in further analysis.

At this point, we created an alignment process that mapped reads to a reference genome and identified the exact genomic loci corresponding to each read. Given that, we did not use primers for amplification of the SARS-CoV-2 genome, and we had to remove all the reads that were mapped with other material sequenced from the swabs. In this study, to remove the contaminating sequences, we considered humans, fungal, and bacterial reference genomes, respectively. To elaborate, we used the “Homo sapiens genome assembly GRCh38 (hg38)” from the Genome Reference Consortium [20] as the human reference genome and all sequences from both fungal and bacterial genome projects from NCBI. Lastly, we used the NCBI SARS-CoV-2 sequence NC_045512.2 [14] as a reference genome for SARS-CoV-2.

For each of these reference genomes, we mapped the reads using the minimap2 (v2.17–r941) tool [21] based on earlier reports that demonstrated the effective performance of this tool with the splice-aware alignment of nanopore direct RNA reads against a reference genome [22]. As suggested by authors who reported the performance of minimap2, we set the parameter k = 13 to increase the sensitivity and to map noisy nanopore direct RNA seq. reads.

Next, we extracted both unmapped reads and reads with mapping quality lower than 10 using the “view” utility in the samtools (v1.7) library [23] for each reference genome except for that of SARS-CoV-2.

Resultantly, we obtained a subset of the sequenced long reads that did not map with the genomes of humans, fungi, or bacteria.

Lastly, we mapped the remaining reads against the SARS-CoV-2 reference genome using minimap2 tools with the same parameters used earlier. We successively analyzed the results of the mapping process using the BCFtools (v1.9) library [24], a set of utilities for variant calling. In particular, we first used the mpileup tool [16] to generate a summary of the coverage of the mapped reads on the SARS-CoV-2 reference genome at single base-pair resolution, followed by the call tool [25] for generating calls. We set these tools to perform as the standard consensus caller with only the variant sites returned. The results of this pipeline were enlisted as SARS-CoV-2 variants detected in the sequenced samples from the swabs.

### 2.3. Calculation of Mutation Frequency

Sanger sequencing. The samples were sequenced using the BigDye™ Terminator v3.1 Cycle Sequencing Kit (Life Technologies, Carisbad, GA, USA; Cat. No. 4337455) with an ABI PRISM 3100 Genetic Analyzer upgraded to the Applied Biosystems^®^ 3130xl System (Life Technologies; Cat. No. 4359571). To enrich the identified mutation region in the N gene, we adopted a nested PCR approach using outer and inner primer sets. Both primer sets are reported in Appendix A.

Two-step RT-qPCR. The ten sequenced samples and the two uninfected control RNAs were retrotranscribed using the ExcelRT™ Reverse Transcription kit (SMOBIO, Taiwan, China; Cat. No. RP1300). First, 16 µL of each sample was mixed with 2 µL of 50 µM of Oligo (dT)20 primer and 2 µL of dNTP Mix (10 mM each). The mixes were incubated at 70 °C for 5 min and placed on ice for at least 1 min. Next, 8 µL of 5× RT buffer (DTT), 8 µL of DEPC-treated water, 2 µL of RNAok™ Rnase Inhibitor, and 2 µL of ExcelRT™ Reverse Transcriptase was added to each sample. The reactions were incubated at 25 °C for 10 min, at 44 °C for 50 min, and at 85 °C for 5 min, and were subsequently held at 4 °C in Mastercycler Pro S (Eppendorf, Hamburg, Germany; Order No. 6325EJ921257).

Next, to amplify the region containing the NC_045512:c.608_609_610delinsAAC mutation, we designed two primer sets to distinguish between a wild type (WT) and a mutated viral strain. The two sets used the same reverse primer, with divergence at the last three bases at the 3′ end of the forward primer. In fact, one forward primer was specific to the mutated sequence, whereas the other recognized the reference sequence.

Furthermore, we used two other sets as controls. One set consisted of the primers N2 from the 2019-Novel Coronavirus (2019-nCoV) Real-Time rRT-PCR Panel (Centers for Disease Control and Prevention (CDC) Atlanta, GA 30333, USA), which was used to confirm the presence of the N gene target (Appendix A). The second set consisted of a custom-specific primer for human GAPDH cDNA, which was used as an endogen control for the PCR (Appendix A). All primer sequences are listed in Appendix A.

Moreover, a plasmid vector carrying a synthetic reference SARS-CoV-2 N gene (Origene, Rockville, MD, USA; Cat. No. VC202563) was used as a reference control.

First, both analysis samples and uninfected control cDNA samples were diluted at a 1:2 ratio, while the reference control was concentrated to 0.8 ng/µL. Next, 4 µL of each sample in 20 µL of reaction mixture was assayed by qPCR using the Quantinova SYBR Green PCR kit (Qiagen, Hilden, Germany; Cat. No./ID: 208052) according to the manufacturer’s instructions in a Rotor-Gene Q 2plex HRM Thermocycler (Qiagen, Hilden, Germany; Cat. No./ID: 9001861). The following PCR thermal profile was used: 95 °C, 2 min; 95 °C, 5 s; and 57 °C, 10 s for 35 cycles.

## 3. Results

To characterize the SARS-CoV-2 mutational landscape, we used RNA extracted from samples collected from five male and five female patients who tested positive for SARS-CoV-2 in the swab test, with the corresponding Ct values ranging from 18 to 24. First, we assessed the concentrations and the fragmentation levels in the samples. The overall samples had significantly low concentrations (in the order of pg/µL) and high fragmentation levels with an RIN index ranging from 2.6 to 2.1. Next, we prepared Pools A and B from the samples. We prepared the former to increase the abundance of sequencing output and the latter to improve the heterogeneity in the data.

We set up each pool at a final volume of 10 µL and prepared the libraries for direct RNA seq. After each purification step, the concentrations were 1.2 ng/µL and 0.6 ng/µL in the initial eluates and 0.9 ng/µL and 0.3 ng/µL in the final eluates at the presequencing stage for Pools A and B, respectively. Although the library concentrations were 200 times less than that required for the protocol, we opted to load the libraries and to launch MinKNOW to test the suitability of MinION. To perform a single MinION sequencing run, we loaded Library A in the flow cell at the beginning of the run with 1575 active pores enabled, while Library B was loaded when 1260 active pores remained active. Because the final library concentration was suboptimal, we observed that only a limited number of pores were active at a time, as shown in the MinKNOW duty time plot in Figure 2.

Next, we launched the base calling of the reads using *Guppy* tools and obtained a set of 397,465 reads in the Fastq format file.

As stated previously, the overall quality of reads is affected by the adopted sequencing technique, i.e., the direct RNA seq. of samples derived from the swabs. In this experiment, the average read length was lower than that of standard long reads, and the quality of certain reads was below conventional levels. Therefore, we preferred to discard the reads that had low quality (<8), as well as a short length (<500 nt). We obtained a subset of 20,940 good-quality reads.

At this point, we aimed to identify and remove sequences from contaminant species contained in the swab; for this, we used the *minimap2* tool to filter out reads that mapped with sequences from human, fungal, and bacterial genomes.

Using this method, we filtered 97.63% of the reads; details of the composition of these reads are provided in Appendix A. Lastly, we attempted to map the remaining 2.37% of reads with the SARS-CoV-2 reference genome and observed that 10.89% of the reads (i.e., 54 reads) were mapped. These reads did not cover the entire SARS-CoV-2 reference genome; however, these were sufficient for mapping the N gene.

Next, we analyzed the coverage of the mapped reads and the variant calling related to the N gene using the *SAMtools* and *BCFtools* libraries.

The analysis revealed the existence of the following mutation region with a quality score greater than 94.99: NC_045512:c.28881_28882_28883delinsAAC. The N gene sequence generated is available in the GenBank database and can be accessed at: https://www.ncbi.nlm.nih.gov/nuccore/MT457389 (accessed on 12 May 2020).

Figure 3 shows the results of the alignment phase with the Integrative Genomic Viewer (IGV) (v2.8.2) tool. The manner in which the reads mapped considerably well with the region of interest in the SARS-CoV-2 reference genome is apparent, and the coordinates where the mutation appears are clearly visible.

To determine the frequency of mutation in the samples, we adopted two different approaches: Sanger sequencing and qPCR. Using the former approach, we amplified the region that contained the mutation prior to sequencing it. As shown in Figure 4, all samples (100%) contained the NC_045512:c.28881_28882_28883delinsAAC mutation.

In the qPCR assay, we used custom primer sets designed to distinguish between WT and mutated sequences and another pair set. We compared two uninfected controls and the synthetic SARS-CoV-2 N cDNA, which was similar to the positive control of the reference genotype. All the samples yielded positive results with the primer set specified for the mutated sequence (Appendix A) and negative for the WT sequence (Appendix A), in contrast to that for the positive reference control, consistent with the results of the Sanger sequencing experiment. The uninfected samples tested negative for both mutated and WT sets, which confirmed the absence of a possible off-target from the host.

At last, we investigated whether findings from other studies on SARS-CoV-2 were consistent with our findings. To this end, we considered sequence variations from the China National Center for Bioinformation (CNCB) portal [26]. CNCB is the largest, daily-updated, publicly available SARS-CoV-2 genome variation repository. It contains more than 21,000 high-quality, human-hosted, complete SARS-CoV-2 genomes (last download on 12 June, 2020) from several countries worldwide. We retrieved this dataset and prepared a script to check for the presence of the identified mutation region (NC_045512:c.28881_28882_28883delinsAAC) in the sequences of the N gene present in the CNCB dataset. As shown in Figure 5A, we considered three sets: (1) cases that only contained identified mutations, (2) cases that also contained identified mutations, and (3) cases that did not contain the identified mutations. We found that the same N gene sequence we reported was reported in approximately 18% of COVID-19 cases, and we took into account the distribution of these cases over time.

As shown in Figure 5B, we reported the presence of the mutation in cases treated per week from February to June and noticed that the majority of cases were reported between the second week of March and the fourth week of April. This period coincided with that of swab collection from patients.

## 4. Discussion

Nowadays, next-generation sequencing technologies may be used to analyze whole RNA transcriptomes of complex organisms and whole DNA or RNA genomes of microorganisms, within a short duration and at an affordable cost [3]. The coverage and depth of the sequencing reads are remarkably influenced by the harvest of the sample and on the matrix and protocol used for extraction [27,28]. Protocols used for the sequencing RNA library preparation often require at least 100 ng total RNA, and unfortunately, some biological matrices, such as an oropharyngeal swab, do not satisfy this request. In this case, in order to comply with guidelines and to increase the likelihood to obtain sufficient RNA amount in terms of the coverage and depth for the robustness of the sequencing data [29], RNA enrichment through amplification reactions of the polymerase enzymes is necessary [3]. However, the nature of the polymerases is to introduce errors of the read and/or other synthesis artifacts [30]. Thus, for example, during the synthesis of cDNA, the development of artificial RNA–RNA chimeras is possible, which can generate noise during the assembly of the reads [11]. Other research has used several library preparation protocols suitable for different NGS platforms, including nanopore, reporting several drawbacks of cDNA sequencing. For instance, amplicons of viral transcriptomes, which were investigated using sequencing-based nanopores, showed bias from reverse transcription and amplification [31,32,33]. This aspect makes the protocols of cDNA sequencing less suitable for specific analyses, such as the identification of nucleotide variants [4,5]. 

Oxford Nanopore protocols SARS-CoV2 PCR tailing based are suggested to sequence the coronavirus genome directly from nasopharyngeal or oropharyngeal swabs. In order to cover the whole genome, these protocols, exploiting the technology based on an array of nanopores, use a multiplex primers scheme , which generate longer amplicons and tend to produce close-to-finished genomes more quickly than short reads based sequencing platforms [9]. Moreover, despite a single nanopore long read can have a more relatively higher error rate than short read, studies that compare the genome assembly derived from both technologies demonstrated that through Nanopore it is possible to achieve highly accurate consensus single nucleotide variant (SNV) calling with >99% sensitivity and >99% precision with a minimum of about 60-fold coverage [9].

On other hand, these protocols produce a higher percentage of reads that are not analyzable by the bioinformatic pipelines, likely due to a combination of fragmentation of synthesized molecules and prematurely aborted molecules during sequencing [9]. Brejová et coworkers dissected the genome assembly generated by sequencing of amplicons derived from samples of the same biological material using schemes of primers set, which produced long reads of 400-bp, 2-kb, and 2.5 kb. These experiments highlighted that the protocols based on more long amplicons generated a higher percentage of failed reads compared to the sequencing of shorter amplicons, after the quality filters used from Artic pipeline. The majority of failed reads was due to the low quality or incompleteness, often leading to the inability to recognize either barcodes. Moreover, the authors reported that the coverage of some amplicons widely varied, with regions which showed lower depth or were completely missing, in that cases where mutations involved sites overlapping PCR primers.

In this study, we showed as direct-RNA seq. on an array of nanopores can help to identify the native sequences of the genome of microorganisms at RNA, which could be present in clinical samples such as nasopharyngeal or oropharyngeal swabs. This protocol has the advantage of directly detecting the ribonucleobases passing through the pores without RT or PCR amplification, and this approach is free of possible biases or mis-amplifications introduced during such steps. This direct sequencing also simplifies the gene annotation process, allowing so the identification of more complex or novel transcript isoforms genome wide [34,35], and better able to differentiate transcript haplotypes [36]. Moreover, the direct-RNA seq. is more suitable than the conventional RNA-seq method for the study of RNA viral genomes, because they show several challenges such as multiple reading frames, anti-sense gene locations, inefficient termination signals, and complex splice forms, and gene annotation [37]. In point of fact, the nanopore direct-RNA seq. has been used to study the transcriptome of DNA viruses such as HSV [37] and to completely sequenced the genome of influenza A in its original form [38].

Given the increasing prevalence of SARS-CoV-2 cases confirmed in several countries and the resulting need for unbiased sequencing approaches, our research group performed an Oxford Nanopore direct-RNA seq. experiment on oropharyngeal swab samples of Italian patients bearing SARS-CoV-2 from the Palermo area (Sicily).

A similar approach was described on RNA of good quality extracted from cell cultures [11]. 

In our study, we identified COVID-19 sequences from both high fragmentation (RIN index ranging from 2.6 to 2.1) and very low concentration RNA (in the order of pg/µL), which is directly extracted from the routine oropharyngeal swab.

Therefore, in order to improve the abundance and the heterogeneity of the sequencing reads, we chose to sequence two libraries, A and B, in the same experiment, as previously described.

Despite these limitations, through an appropriate bioinformatics pipeline used to analyze the Fast5 sequencing data, we detected in all samples used in this study the presence of the NC_045512:c.28881_28882_28883delinsAAC mutation, which is a set of mutations in the SARS-CoV2 phylogenetics already described in the literature.

In order to confirm the robustness of the obtained data, we compared the in silico results using Sanger sequencing and qPCR assays, as described in the Results section (Section 3). These assays identified in all samples the presence of the NC_045512:c.28881_28882_28883delinsAAC mutation, as well as the results of the nanopore direct RNA seq., confirming this method.

## 5. Conclusions

The nanopore direct RNA seq. protocol allows carrying out true native RNA sequencing, without PCR request, to explore attributes of native RNA such as modified bases and to remove bias introduced from RT or PCR. Moreover, it might be recommended in the occurrence of RNAs that are difficult to reverse transcribe [5,11].

Additionally, it could be used for the direct sequencing of the SARS-CoV2 RNA genome, as well as other RNA viral genomes, i.e., in HIV viruses, and it could be also considered advantageous for both surveillance and epidemiologic studies.

## Figures and Tables

**Figure 1 life-12-00069-f001:**
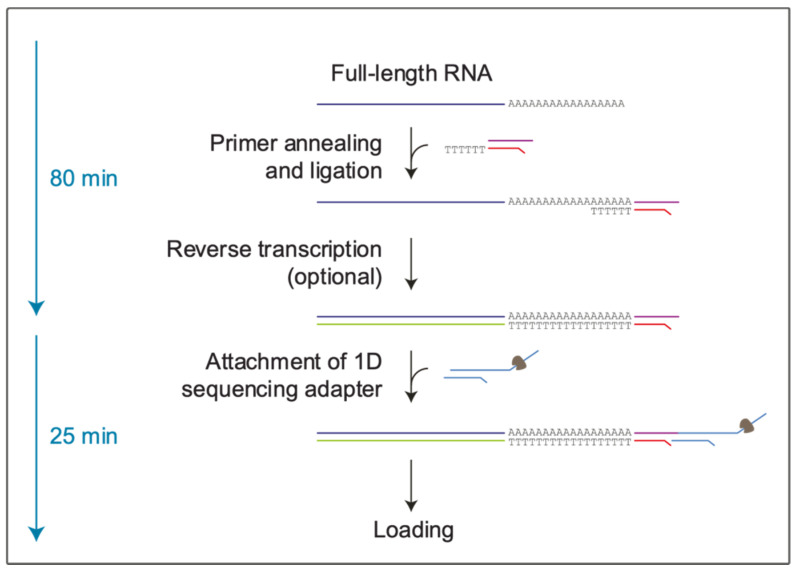
Representative scheme of the direct RNA seq. protocol (SQK-RNA002, Nanopore Technologies). dsRNA-DNA hybrids are synthesized during the library preparation. Next, the adapters needed for sequencing through nanopores are added at the 3′ poly A end, with the motor protein bound in 3′ to 5′ direction. Thereby, only the RNA strand will be sequenced, whereas the cDNA strand will be excluded.

**Figure 2 life-12-00069-f002:**
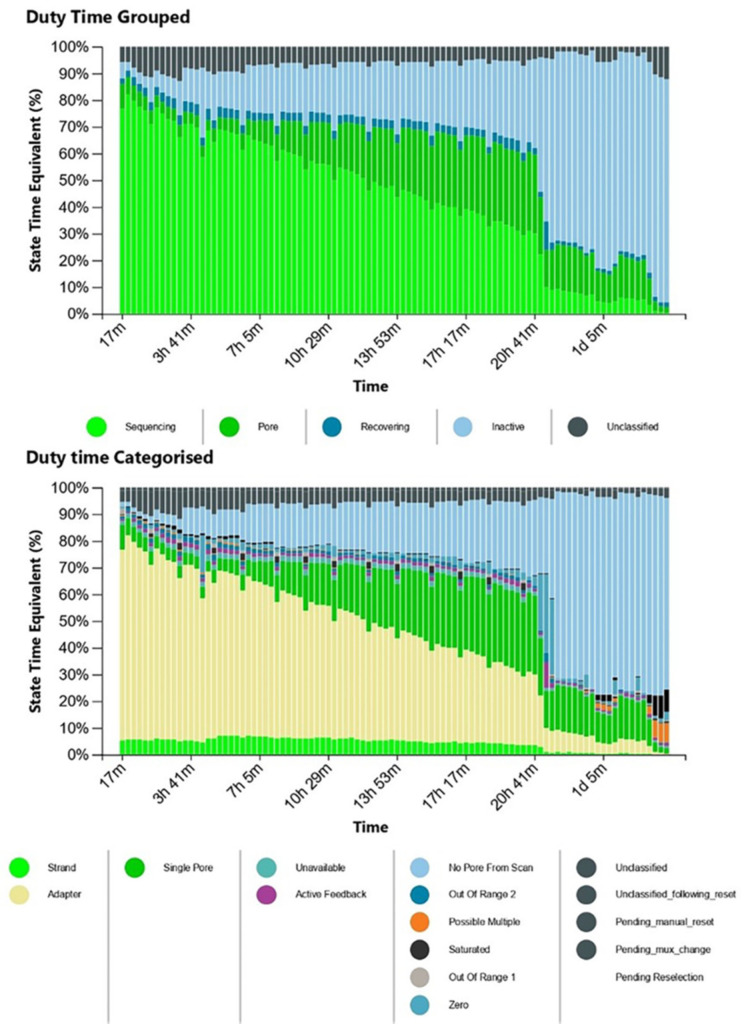
The MinKNOW duty time plots depict the sum of total channel activity within a particular period of time. The number of sequencing pores decrease over time. In the categorized plot, the imbalance of adapters is apparent.

**Figure 3 life-12-00069-f003:**
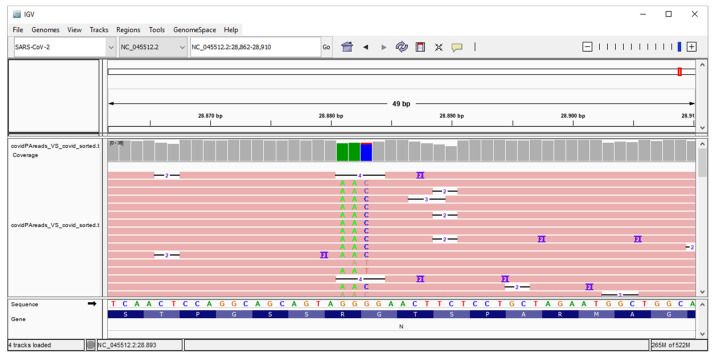
Genome IGV analysis that shows the frequency of the identified NC_045512:c.28881_28882_28883delinsAAC mutation in the reads from sequenced samples.

**Figure 4 life-12-00069-f004:**
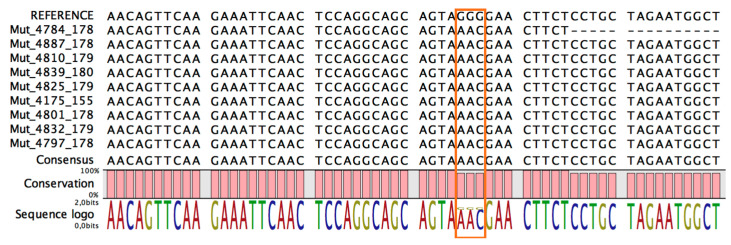
The sequences obtained by Sanger sequencing of RNA extracted from swabs that tested positive for SARS-CoV-2 infection. The rectangle covers coordinates from 28,881 to 28,883 for underlines, as all samples contained the NC_045512:c.28881_28882_28883delinsAAC mutation.

**Figure 5 life-12-00069-f005:**
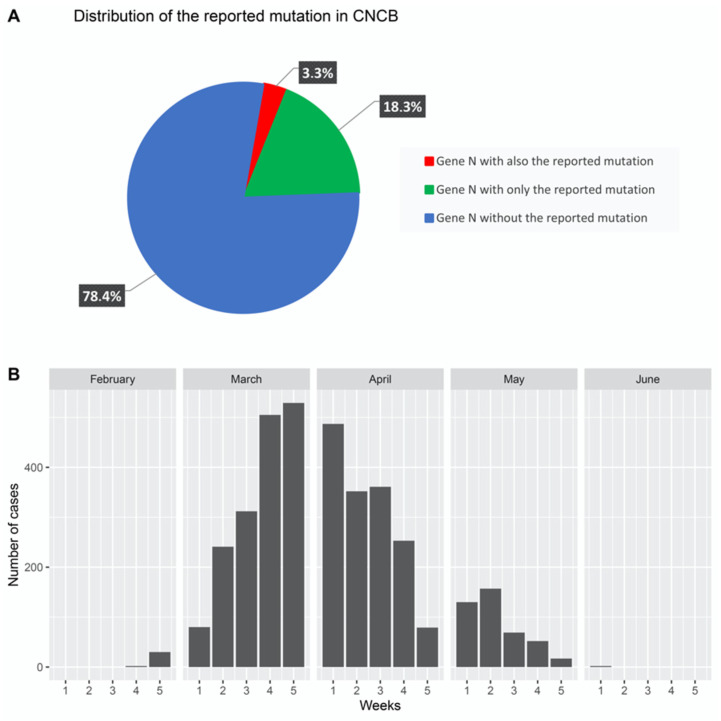
Identification of the reported mutation in the CNCB database. (**A**) Strains isolated from 18.3% of COVID-19 cases (over 21,000 complete genomes sequenced from isolated strains) contained the reported mutation. (**B**) Plot depicting the distribution of these cases from the end of February to the beginning of June.

## Data Availability

The data that support the findings of this study are available from the corresponding author upon reasonable request.

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
