# Peer review of "Direct RNA Nanopore Sequencing of SARS-CoV-2 Extracted from Critical Material from Swabs"

_life, 2022, doi:10.3390/life12010069_

Round 1

Reviewer 1 Report

The authors used the direct RNA nanopore sequencing of SARS-CoV-2 extracted from the biological sample with the third-generation sequencing technology, it has many advantages over others sequencing, and which is very interesting to readers. however, there are some problems to be solved before consideration for publication. (1) there is a lack of the patient's permission document description for the sample use;(2) The description of experimental research in scientific research papers is objective, and the expression in the manuscript needs to be modified to passive; (3) Lack of uniformity in the format of references.

Author Response

Dear Reviewer,
Thanks for your suggestions.
Following are present the  point-by-point responses: 

Q: There is a lack of the patient's permission document description for the sample use.

R: We agreed with the reviewer, and then we added the "Informed Consent Statement" Section at the end of the manuscript, according to the journal template.

Q: The description of experimental research in scientific research papers is objective, and the expression in the manuscript needs to be modified to passive

R: We thank the reviewer for this suggestion, but the manuscript was edited before submission by an external editing service that indicated that we must put the majority of our sentences in the active voice.

Q: Lack of uniformity in the format of references

R: According to the reviewer's suggestion, we checked the format of all the references and fixed those that were not compliant with the MDPI referencing style.

Reviewer 2 Report

This manuscript describes the use of direct RNA sequencing on an Oxford Nanopore platform to identify SARS-CoV-2 variants in oropharyngeal swabs. Despite low yield and high fragmentation of source material, the authors were able to identify a common variant mutation in a limited portion of the CoV-2 genomes corresponding to the N gene. Variants detected by direct RNA sequencing were validated using "gold standard" cDNA sequencing and RT-qPCR. The methods are very well described and the results are presented accurately and with clarity, including appropriate consideration of technical shortcomings. What is missing is any description or discussion of what differentiates this study from the numerous publications that use the Oxford Nanopore platform to sequence and identify CoV-2 genomic features and variants. Given that dozens (if not hundreds) of similar publications are already available, it is critical that the authors highlight what is unique in the current study and make an effort to describe and summarize the previous studies. 

Author Response

Dear Reviewer,
Thanks for your suggestions.
Following are present the  point-by-point responses:

Q: What is missing is any description or discussion of what differentiates this study from the numerous publications that use the Oxford Nanopore platform to sequence and identify CoV-2 genomic features and variants. Given that dozens (if not hundreds) of similar publications are already available, it is critical that the authors highlight what is unique in the current study and make an effort to describe and summarize the previous studies. 

R:  We thank the reviewer for this question because it represents a central point of our investigations. For this purpose, we added in the Introduction and in discussion sections other sentences which highlight the distinctiveness of this study and summarize the previous studies.

Sincerly

Reviewer 3 Report

This is an interesting study on the use of new tools and techniques to investigate SARS-CoV-2 genome. Sincere compliments for your work.

Please specify the novelty of your job with regard to the text that can be found here:

https://www.medrxiv.org/content/10.1101/2020.12.21.20191346v1.full-text  

  • In my opinion, introduction should be improve adding more references.
  • Line 60 “which has been described in literature” lacks references, instead.
  • In line 76, it is unclear to me the reason why you did analyze brain biopsy specimens. The experimental design is here puzzling and should be better explained: for example, I would have expected the use of negative swabs as control.
  • Figure 1 is unclear and should be improved in image definition
  • In results, lines 216-287, you have to delete any reference to material and methods, that I can find in many rows of the paragraph. I can also find literature reference in line 245, 252-253. Results have to be written with your experimental results, without referring to the protocols.
  • Figure 2 appears cut in the left and unclear.
  • In line 309, Figure 4 and 5 appear to me as a results and have to be included in Results.
  • Figure 3 and Figure 4 appear unclear in image definition.
  • Figure 5 contains graphics of poor quality. Please improve data plotting and in particular improve the pie-chart.
  • In line 291 please substitute harvest with sampling, for example.
  • In line 382-383, there is a reference to supplementary materials code, that is not working. Please have a check.
  • Please complete the paper with conclusions: I can only read the discussion in paragraph 4. Line 288.

In conclusion, the authors should review and redefine Results and Discussion and finally adding Conclusions.

Author Response

Dear Reviewer,
Thanks for your suggestions.
Following are present the  point-by-point responses: 

Q1: Please specify the novelty of your job with regard to the text that can be found here: https://www.medrxiv.org/content/10.1101/2020.12.21.20191346v1.full-text

R1: As noticed by the reviewer, there is no difference between the first version of this manuscript and the unreviewed version archived on the "medrxiv" website. Of course, this revised version includes some significant differences, given by the introduction of valuable improvement provided by reviewers' suggestions.

Q2: In my opinion, introduction should be improve adding more references.

R2: We thank the reviewer for this comment. We added new references at the introduction of the manuscript

Q3: Line 60 "which has been described in literature" lacks references, instead.

R3: We added the missing reference.

Q4: In line 76, it is unclear to me the reason why you did analyze brain biopsy specimens. The experimental design is here puzzling and should be better explained: for example, I would have expected the use of negative swabs as control.

R4: We better explain the motivation of using brain biopsy samples in the 2.1 sub-section. We added the following sentences: “In order to assay the suitability of our custom primers and to avoid possible PCR artifacts derived from high fragmented and very low concentrated RNA, such as that extracted from the swab, we used as negative controls high-quality RNA extracted from brain biopsies of two patients who underwent surgery before of the beginning of the epidemic in Italy”.

Q5: Figure 1 is unclear and should be improved in image definition

R5: Done.

Q6: In results, lines 216-287, you have to delete any reference to material and methods, that I can find in many rows of the paragraph. I can also find literature reference in line 245, 252-253. Results have to be written with your experimental results, without referring to the protocols.

R6: Thank you. We deleted all the references to the Material and Methods section and re-arranged the manuscript according to the reviewer suggestion.

Q7: Figure 2 appears cut in the left and unclear.

R7: We made a new screenshot for this figure.

Q8: In line 309, Figure 4 and 5 appear to me as a results and have to be included in Results.

R8: We agree with the reviewer. Both figures are now in the Results section.

Q9: Figure 3 and Figure 4 appear unclear in image definition. Figure 5 contains graphics of poor quality. Please improve data plotting and in particular improve the pie-chart.

R9: We thank the reviewer for this suggestion. We made a re-caption of both Figure 3 and Figure 4. Also, we improved the quality of the plot in Figure 5.

Q10: In line 291 please substitute harvest with sampling, for example.

R:10 Done.

Q11: In line 382-383, there is a reference to supplementary materials code, that is not working. Please have a check.

R11: We submitted the supplementary material code together with the main manuscript. At this moment, we don't know the final URL. It will be created by MDPI if the manuscript will be accepted for publication.

Q12: Please complete the paper with conclusions: I can only read the discussion in paragraph 4. Line 288.

R12: We thank the reviewer for this comment. We added a Conclusion Section to the manuscript.

Sincerly, 

Round 2

Reviewer 2 Report

The impact of the current study is now much more evident and discussion of similar studies is improved. The authors have adequately addressed my initial concerns.

Author Response

Thanks, Reviewer for your advice

I want to take this opportunity to wish you a Merry Christmas

Sincerely

Davide Vacca